# Numerical stability of DeepGOPlus inference

**Inés Gonzalez Pepe**[1]*, **Yohan Chatelain**[1], **Gregory Kiar**[2], **Tristan Glatard**[1]

**1** Department of Computer Science and Software Engineering, Concordia University, Montreal, Qc, Canada,
**2** Computational Neuroimaging Laboratory, Child Mind Institute, New York, NY, United States of America

* i_gon@encs.concordia.ca

**Data Availability Statement:** The data used in this paper can be found at http://deepgoplus.bio2vec.net/data labelled as version 1.0.6. This dataset was originally used in the DeepGOPlus paper to compare the model to other models in the fields,

## Abstract

Convolutional neural networks (CNNs) are currently among the most widely-used deep neural network (DNN) architectures available and achieve state-of-the-art performance for many problems. Originally applied to computer vision tasks, CNNs work well with any data with a spatial relationship, besides images, and have been applied to different fields. However, recent works have highlighted numerical stability challenges in DNNs, which also relates to their known sensitivity to noise injection. These challenges can jeopardise their performance and reliability. This paper investigates DeepGOPlus, a CNN that predicts protein function. DeepGOPlus has achieved state-of-the-art performance and can successfully take advantage and annotate the abounding protein sequences emerging in proteomics. We determine the numerical stability of the model's inference stage by quantifying the numerical uncertainty resulting from perturbations of the underlying floating-point data. In addition, we explore the opportunity to use reduced-precision floating point formats for DeepGOPlus inference, to reduce memory consumption and latency. This is achieved by instrumenting DeepGOPlus' execution using Monte Carlo Arithmetic, a technique that experimentally quantifies floating point operation errors and VPREC, a tool that emulates results with customizable floating point precision formats. Focus is placed on the inference stage as it is the primary deliverable of the DeepGOPlus model, widely applicable across different environments. All in all, our results show that although the DeepGOPlus CNN is very stable numerically, it can only be selectively implemented with lower-precision floating-point formats. We conclude that predictions obtained from the pre-trained DeepGOPlus model are very reliable numerically, and use existing floating-point formats efficiently.

## Introduction

Recent advances in the field of proteomics have resulted in an abundance of protein sequences that cannot feasibly be identified by traditional experimental means, thus leading to innovations in computational protein function detection methods. While many proteins are being discovered, little is known about them or their function. Proteins play important roles within organisms as catalysts and drivers of many biochemical reactions, but in order to understand these reactions, the proteins' functions must be known. Understanding the functions of these newly discovered proteins is pivotal, particularly in the context of drug discovery, where manipulating protein behavior through tailored molecules is a core strategy. Leveraging a

but the dataset has since been updated since the original publication of the DeepGOPlus paper to reflect new data releases. The dataset is composed of the Gene Ontology released on 2021-10-26 and the SwissProt data version 2021_04. The data consists of experimental annotations collected before a certain date as the training set and confirmed experimental annotations discovered later as the test set. The data generated by this study is available on Zenodo: 10.5281/zenodo. 8371238.

**Funding:** The author(s) received no specific funding for this work.

**Competing interests:** The authors have declared that no competing interests exist.

protein's sequence, which is often the most readily available information, allows for the optimization of drug design. Automating this process can allow for the large amounts of data being produced to be leveraged to decrease the time needed to design medication and reduce the trial and error of determining how man-made molecules should interact with proteins [1, 2].

Deep neural networks (DNNs) have become popular approaches for protein function prediction in recent years, due to requiring significantly less time and human effort to achieve good performance relative to existing experimental means [3–10]. Nonetheless, while current DNN models are able to achieve high predictive performance, the numerical reliability of these predictions—defined as the stability with respect to minor numerical perturbations—is unknown and should be evaluated for several reasons. Considering DNNs often operate as "black boxes" for which we lack a full understanding, it is essential to understand their behaviour or at least their numerical properties. With this objective in mind, our aim is to delve into the numerical stability of a DNN, not only to gauge its reliability but also to shed light on its underlying numerical properties. Changing execution environments has also been shown to affect the reproducibility of results in bioinformatics, in part due to numerical instabilities [11], which compromises the ability to reliably form computational predictions across contexts. Moreover, adversarial attacks have shown that perturbing input data with small amounts of noise can drastically impact classification outcomes [12–14], which suggests that minor numerical perturbations in the model itself could similarly impact the predictions.

Furthermore, numerical stability has strong implications on the numerical precision required in model training and inference. In turn, appropriate floating point data formats have a significant impact on both the run-time and memory requirements of models both at training and inference time. In recent years, the use of reduced or mixed precision has yielded substantial improvements to resource requirements for DNNs, including CNNs [15–18]. These efforts do however come back to numerical stability, as reduced precision can only be attempted on a model that is already stable at full precision and must be similarly stable at a lower precision without a drop in performance.

This paper centers its investigation on DeepGOPlus [19], a state-of-the-art CNN model for protein function classification. DeepGOPlus, as a computational approach, offers significant advantages such as the rapid processing of extensive protein sequences compared to traditional experimental methods. Notably, it simplifies user interaction by solely requiring the protein sequence as input, eliminating the need for complex additional information about the protein. While CNN architecture stands as one of the most widely adopted methods for protein function prediction, it is worth acknowledging the success of alternative approaches, including K-Nearest-Neighbors and Logistic Regression [8–10, 20, 21]. Many models in this domain typically incorporate supplementary protein information, such as protein-protein interactions and structural data, to enhance their performance. However, DeepGOPlus was specifically chosen due to being a top performer in the CAFA3 challenge [22], a community-wide assessment of computational protein function prediction methods. Furthermore, the availability of both the DeepGOPlus model and its associated data makes it an ideal choice for this study, facilitating accessibility and further analysis.

This paper aims to (1) determine the stability of DeepGOPlus inference in response to minute numerical perturbations, (2) evaluate the possibility of using reduced numerical precision formats during inference when using DeepGOPlus as illustrated in Fig 1. To achieve the first goal, we study the numerical stability of DeepGOPlus by using Monte Carlo Arithmetic (MCA) [23]. MCA is a stochastic arithmetic technique to empirically evaluate numerical stability by injecting noise into floating point operations and quantifying the resulting error at a given virtual precision. Other approaches to determining numerical stability involve formal

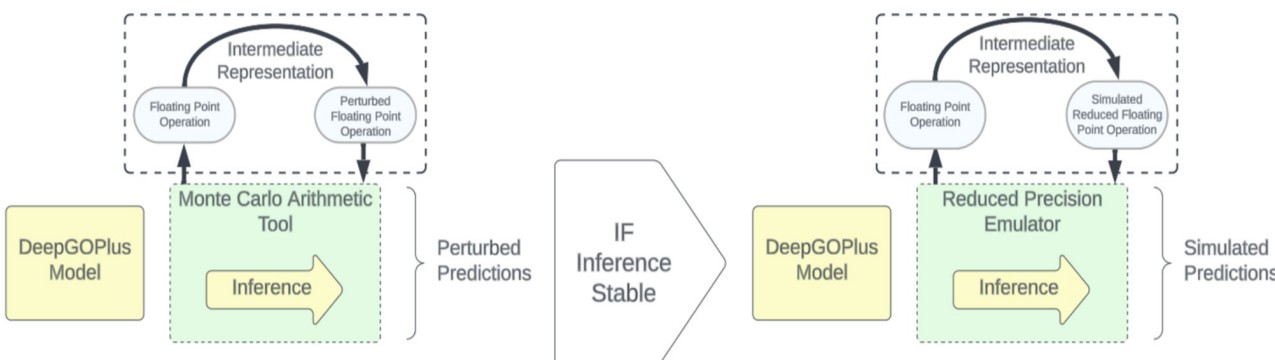

**Fig 1. Illustration of proposed perturbation and simulation of CNN inference methods.**

error analysis and interval arithmetic [24], but unlike MCA they do not scale to large code bases and require modification of the source code. Moreover, MCA is chosen over another stochastic arithmetic technique, Discrete Stochastic Arithmetic (DSA) [25], as DSA is less scalable to large code bases and makes limitative assumptions about the nature of the distribution of computational errors. We apply MCA to the model through the Verificarlo [26] and Verrou [27] software tools, which dynamically replace floating point operations by perturbed ones at program execution time. The perturbations introduced simulate noise typically encountered when running code in different environments subject to library updates, OS and/or hardware changes, and allow for assessing the precision of numerical results. To achieve the second goal, we emulate reduced precision formats with the VPREC tool [28] and observe the corresponding predictive performance. We evaluate reduced precision formats that have existing hardware implementations to increase the practical value of our results. Our contributions can be summarised as follows:

• Quantification of the numerical stability in the DeepGOPlus CNN's inference stage, providing a reliability metric for model users.

• Exploration of the impact of reduced precision on the performance of the DeepGOPlus CNN through simulation.

The next section discusses related work on the numerical stability of DNNs and their use of reduced precision. The methods section introduces the tools mentioned above such as MCA, Verificarlo, Verrou and VPREC, as well as the dataset used for our evaluations and other experimental parameters. The results section presents our evaluation of the numerical stability of DeepGOPlus, as well as the possibility of implementing reduced precision formats in the inference stage.

## Related work

### Numerical stability of DNNs

Prior work has studied DNNs to both identify numerical instabilities and improve their implementation. One such approach is the DeepStability project, a public database of numerical stability issues and solutions in PyTorch and TensorFlow [29]. DeepStability was created by studying the PyTorch and Tensorflow code repositories for issues revolving around numerical stability, which required extensive manual source code evaluation. Monte Carlo Deep Neural

Network Arithmetic (MCDA) [30] is another approach to numerical stability evaluation that performs systematic permutations of networks. MCDA measures the sensitivity of a model in regards to the error introduced by its use of floating-point operations. MCA can be implemented more generally through different tools such as Verrou or Verificarlo and can be scaled up to analyse large code bases such as Tensorflow.

Numerical instabilities can have different origins in DNNs, among them, the algorithmic implementation of the model. Numerical expressions in general can be implemented by different algorithms which may vary in their stability. As a result, ResNet models with forward propagation algorithms more stable in regard to vanishing or exploding gradients [31] and skip connections rendered more robust to adversarial attacks with the implicit Euler method [32] have been proposed. These works differ from the ones mentioned earlier in this section as they propose stable implementations to specific unstable problems while the former propose tools to be used across DNNs to identify numerical unstability.

The importance of investigating numerical stability is highlighted by the impact of adversarial attacks, i.e., minor perturbations of the input data to mislead the model, which have been shown to drastically change classification results in some cases [12]. When expanding the scope of numerical stability of DNNs to include robustness against adversarial attacks, we can see that popular strategies involve detecting attacks, mitigating them through data manipulation or mitigating them through network manipulation [33, 34]. Identifying unstable components of a model falls under mitigating adversarial attacks through network manipulation, but this approach can be used in conjunction with others. Another example of mitigating an attack through network manipulation is adding a defense layer to minimize perturbations [35]. Meanwhile, an example of mitigating the attack through data manipulation is stability training, i.e., exposing the model to noisy or adversarial data in order to render it more robust. The primary difference between these approaches and improving the stability of the algorithmic implementation of the model is that the aforementioned approaches modify the model structure, while addressing the algorithmic implementation modifies the internal workings of the model.

In brief, there exist a variety of approaches to addressing the numerical robustness of DNNs, particularly in response to adversarial attacks. Nonetheless, it is important to note that while a lot of these approaches have been attempted on CNNs, the class of the model studied in this paper, most of the implementations suggested are for image-classification tasks [36, 37] while our study focuses on protein function classification. Applying adversarial attacks to protein strings would most likely involve additional input data transformations [38]. Moreover, adversarial attacks introduces a specific form of noise to the input data to mislead a model, while MCA introduces random noise uniformly throughout a model's floating point operations in order to quantify numerical stability from within.

## Reduced precision for DNNs

DNNs are becoming widely adopted, but one of their limitations is the amount of computing resources required for their training and deployment. The computational complexity of a CNN is often determined by the number of arithmetic operations it does. This is mathematically represented by [39] as:

$$\text{Number of arithmetic operations} = O_h O_w C_o C_i K_h K_W \tag{1}$$

Where $O_h$ and $O_w$ refer to the height and width of the output feature maps, $C_o$ and $C_i$ are the number of output and input channels, and $K_h$ and $K_w$ are the height and width of kernels. However, this is just a general equation as the number of arithmetic operations can be

impacted by pooling layers used in a CNN as well as the algorithmic approach to doing certain arithmetic operations. Nonetheless, while reduced precision does not cut down on the computational complexity of the CNN, it has been shown to decrease the resource consumption of the model, which is why it is of interest in this study. One approach to address this problem is implementing reduced precision formats to replace the standard double-precision (float64) and single-precision (float32) IEEE-754 formats used by default. Benefits of reducing the precision involve a direct decrease in memory usage and an indirect decrease in the energy and time it takes to run an arithmetic operation.

There are many approaches to reducing precision in DNNs. One is to reduce the precision for the entire model life cycle; that is, to restrict the data space being used initially in training, which will cascade to eventual deployment. Another is to employ mixed precision, a technique that selectively reduces the precision in certain operations, while leaving others intact, based on the concept of reducing memory consumption where possible, but ensuring operations such as optimizer and weight adjustment steps do not suffer from reduced precision. Besides only reducing precision, fixed point operations have also been investigated along with floating point operations to improve computing efficiency [40]. However, our study focuses on floating point operations which generally offer a wider range of numbers and a larger dynamic exponent range than fixed point as well as being the default for the DeepGOPlus model.

Using mixed precision usually involves more instrumentation on the part of the user while reducing precision across the board is more likely to cause a drop in performance or in stability [41]. Studies have shown that reducing the precision at the inference stage of the DNN can result in negligible loss in performance, while training the DNN with reduced precision is more difficult due to the importance of preserving the gradients during backpropagation [42]. Nonetheless, many recent papers have shown that comparable performance can be achieved with DNNs trained on 16-bit precision and in some cases with 8-bit precision with either approach [15–18].

## Materials and methods

This section first outlines the principles of Monte Carlo Arithmetic before delving into the tools used to implement it. It then presents reduced precision formats and how to simulate their implementation using the VPREC tool. Finally the dataset, model and experimental setup are explained in detail.

### Use of Monte Carlo arithmetic for numerical uncertainty quantification

To establish the numerical stability of DeepGOPlus, we quantify the numerical uncertainty of the floating point model using Monte Carlo Arithmetic (MCA). We use MCA to inject numerical perturbations into the DeepGOPlus model and quantify the resulting error in the system. MCA leverages randomness to model loss of accuracy inherent to floating point arithmetic due to its finite precision. MCA simulates floating point roundoff and cancellation errors through random perturbations, allowing for the estimation of error distributions from independent random result samples. MCA simulates computations for a given virtual precision using the following perturbation:

$$inexact(x) = x + 2^{e_x - t}\xi \tag{2}$$

where $e_x$ is the exponent in the floating point representation of $x$, $t$ is the virtual precision and $\xi$ is a random uniform variable of $\left(-\frac{1}{2}, \frac{1}{2}\right)$. MCA has three modes that allows perturbation to be introduced in the function input (Precision Bounding—PB), the function output (Random Rouding—RR) or both (full MCA). Random Rounding tracks rounding errors, while Precision

Bounding tracks catastrophic cancellations, and full MCA is a combination of both approaches. Precision Bounding or full MCA which includes PB is not used in this project as it is known to be more invasive and prone to execution failures, which prevents analysis of numerical stability [43]. Only RR, which introduces perturbation in function outputs thus simulating roundoff errors is used in the scope of this project.

$$random\_rounding(x \circ y) = round(inexact(x \circ y)) \tag{3}$$

Noise injected at these virtual precision values reflects typical perturbations caused by changes in the model's environment. These changes may originate from software package changes, different operating systems or different architectures. Therefore, according to Eq 2, for single-precision, $e_x$ ranges between -149 and 127 and $t$ equals 24. We can calculate the relative inputted noise to be $2^{-24} \simeq 10^{-8}$. For double-precision, $e_x$ ranges between -1075 and 1023, $t$ is 53 so the magnitude injected is $2^{e_x - 53}$ corresponding to a relative error of $2^{-53} \simeq 10^{-16}$. The noise is then introduced into the model through Verificarlo and Verrou and 10 iterations are run in order to obtain an adequate sample size to quantify uncertainty.

**Python interpreter MCA instrumentation with Verificarlo.**   Verificarlo [26] is a clang-based compiler that replaces floating point operations by a generic call to one of several configurable floating point models available. MCA is leveraged in the DeepGOPlus model through the use of Fuzzy [44], a collection of software packages compiled with Verificarlo. Fuzzy enables the instrumentation of different libraries by Verificarlo in order to quantify the numerical uncertainty present in code using these libraries. Python is the only existing MCA-instrumented libraries available in Fuzzy that is used in this project.

**Full MCA instrumentation with Verrou.**   Instrumenting Tensorflow as a whole with Verificarlo was unsuccessful since Verificarlo is a clang-based compiler and Tensorflow uses the GCC compiler. Therefore, attempts were made to instrument Tensorflow indirectly through the math libraries used, that is to say either Eigen or MKL. Instrumenting Tensorflow through MKL with Verificarlo did not work since MKL is a closed-source library, which prevents its recompilation by Verificarlo. However, the Eigen library provides the option of outsourcing certain operations to the BLAS and LAPACK libraries which are possible to instrument. Nonetheless, even with the outsourcing to BLAS and LAPACK, the majority of arithmetic operations is done internally by Eigen, therefore attempting to instrument Tensorflow indirectly through Eigen was also unsuccessful.

It is not possible to pass over the instrumentation of Tensorflow as it is the primary software package used to build the DeepGOPlus model as well as the source of most arithmetic operations. Therefore, the Verrou tool was used to instrument the Tensorflow library. Verrou is a tool which also uses Monte Carlo Arithmetic to monitor the accuracy of floating point operations without needing to instrument the source code or recompile it. Verrou is based upon Valgrind [45], a memory-debugging tool which uses dynamic binary instrumentation, i.e. the process of modifying instructions of a binary program while it executes. By default, Verrou instruments the entire executable and produces perturbed results, but libraries can be excluded from instrumentation by the user whenever necessary, such as instrumenting solely Tensorflow. Moreover, while still implementing MCA, it uses its own backend also called Verrou to introduce RR perturbations.

## Use of significant digits as numerical uncertainty quantification metric

Once the MCA samples are obtained, we quantify the uncertainty by calculating the number of significant digits as well as the standard deviation of the classification metrics and class probabilities across the samples. The method used to calculate significant digits also provides

confidence intervals at varying degrees of confidence and probability [46]. These confidence intervals allow for the number of significant digits to be calculated at precise levels of confidence and can inform the user on how many samples should be used in order to achieve high confidence for both the non-parametric and Centred Normality Hypothesis cases. However, we only use the non-parametric case since we can not assume that the output samples follow the Centred Normality Hypothesis.

For a general distribution, computing significant digits requires a bit's significance $S_i^k$ where $k$ is the position of the bit in the mantissa and $X_i$ is a computed MCA sample ($i \leq n$). Given $Z_i = X_i - x_{\text{IEEE}}$, where $x_{\text{IEEE}}$ is the unperturbed result computed without MCA, we can define the bit as significant if the absolute value of $Z_i$ is less than $2^{-k}$.

$$S_i^k = \mathbb{1}_{|Z_i| < 2^{-k}} \qquad (4)$$

Then, to calculate the number of significant bits across samples, $\hat{s}_b$, we determine the maximal index $k$ for which the first $k$ bits of all $n$ sampled results coincide with the reference, which is determined by:

$$\hat{s}_b = \max\{k \in \{1, 2, ..., 53\} \text{such that} \, \forall i \in \{1, 2, ..., n\}, \; S_i^k = \mathbb{1}\} \qquad (5)$$

The number of significant bits measured for a given variable ranges from 0 to 53 bits for double-precision numbers and from 0 to 24 bits for single-precision numbers. A value of 0 significant bits means that the variable has no information while a value reaching the maximal range means that the variable has maximal information given the floating-point format used. The difference between the maximal value and the achieved value quantifies the information loss resulting from numerical instabilities in the model.

To facilitate interpretation, significant bits can be converted to significant digits if necessary. The maximal number of significant digits for double-precision numbers is 15.95 given $53\log_{10}(2)$ and the maximal number of significant digits for single-precision numbers is 7.23 given $24\log_{10}(2)$.

The method used to estimate significant bits also provides a confidence interval on the estimation given the number of MCA samples used. In this study, we chose a sample size of 10, which corresponds to a confidence of 0.80 successful significant digit calculation with 0.85 probability of this value of significant digit occurring.

This method was chosen because of the confidence intervals it provides and because it is more reliable than other stochastic arithmetic methods that also calculate significant digits such as the formulas suggested in the original MCA paper [23] or in the CESTAC (Controle et Estimation Stochastique des Arrondis de Calculs) paper [25].

## Use of VPREc backend for reduced precision emulation

The VPREC backend [28] is used to emulate reduced numerical formats and the results they would produce, thus allowing for an evaluation of how DeepGOPlus functions at lower precisions. VPREC was developed by the Verificarlo team originally for Verificarlo but was made compatible as a backend for Verrou as an additional contribution of this project. It can simulate any floating point format within the double-precision format by modifying the length of the exponent and the mantissa, referred to as precision. VPREC emulates results obtained at customised numerical formats by performing the operation in double-precision and rounding the result to the custom precision and exponent.

The VPREC backend is used to simulate the reduced precision formats of IEEE-754 single-precision [47], IEEE-754 float16 [47], bfloat16 [48] and Microsoft's bfloat8 [49]. These are popular formats that are already implemented in hardware and have already been used to

**Table 1. Numerical formats specifications.** The implicit bit is excluded from the number of bits required for precision, due to normalized numbers always setting this bit to 1.

| Format | Precision (bits) | Exponent (bits) |
|---|---|---|
| float64 | 52 | 11 |
| float32 | 23 | 8 |
| float16 | 10 | 5 |
| bfloat16 | 7 | 8 |
| bfloat8 | 2 | 5 |

reduce precision while maintaining performance and accuracy in different code applications, including in some cases, deep learning algorithms [18]. Table 1 displays the exact precision and exponent range used to simulate each format in VPREC. In addition to these formats, precision values between the range of 52 and 2 and exponent values between the range of 5 and 11 are simulated using VPREC. Despite lacking hardware implementation, this sweep is done to determine the specific point to which DeepGOPlus inference can be reduced without its performance dropping.

VPREC was chosen due to the ease with which it could be applied and because it does not require any modification of the source code unlike other reduced precision techniques. VPREC is utilised as an exploratory technique to investigate the possibility of implementing reduced formats. By emulating reduced formats, VPREC allows for us to simulate the model's predictions for each format, but it does not simulate the overall performance of the model in terms of computational efficiency or memory consumption.

Such metrics are not analysed in our study as they would not properly reflect the model's performance at actual reduced precision. In order to physically re-implement DeepGOPlus at reduced precision, other methods should be explored [50], although most DNNs have required source code modifications or recompilation to adjust their precision. Nonetheless, existing work on reduced precision has demonstrated that reducing single-precision to float16 using mixed precision achieved a speedup of 2–6x, while maintaining equivalent performance across a variety of DNNs [15].

## DeepGOPlus model

DeepGOPlus utilises the Gene Ontology (GO) [51], a hierarchical classification that separates protein functions into annotations that each fall under three super-classes; Biological Process (BP), Molecular Function (MF) and Cellular Component (CC), in order to predict the protein's function by inferring the protein's associated Gene Ontology annotations. Unlike other models, DeepGOPlus only takes as input the protein sequence and does not require additional features such as protein-protein interactions, contact maps, protein structure, etc. This additional information about the protein can be difficult to calculate especially when trying to predict the function of recently discovered proteins little is known about. DeepGOPlus is one of the three best predictors in the Cellular Component class and the second best performing method in the Biological Process and Molecular Function evaluations compared to other models based on the evaluations from the CAFA3 challenge [52].

As illustrated in Fig 2, the model is composed of one convolutional layer composed of 16 one-dimensional convolutional layer components in parallel with different filter lengths that are then sent to their respective max pooling layer before coming together in a fully connected layer which returns a vector with a specific probability score associated with each GO annotation. For the intermediate layers, a ReLU activation function is used, while a Sigmoid

**Fig 2. Architecture of DeepGOPlus CNN (extracted from [19]).**

activation function is used for the final classification layer. These protein function class probabilities are then individually combined using a weighted sum with their counterpart probabilities which are obtained using a different protein function predictor called Diamond [53]. The model is trained on data from the CAFA3 challenge, a community experiment to compare protein prediction methods, and from SwissProt [54], a protein function database.

A model already trained on the v1.0.6 data is made available, as detailed in the previous section, which simplifies the task of studying the inference step. Moreover, we excluded the Diamond tool to determine the instability originating purely from the CNN.

## Experimental setup

The images used to execute the experiments are available on Docker Hub, with instructions on how to find the image on this study's GitHub README and the generated results are available on Zenodo. The model is run in Singularity containers over 10 MCA iterations in order to obtain perturbed samples from which to get a measure of the instability within the model. The container with the instrumented Python interpreter was perturbed using the Fuzzy software tool while Tensorflow and the entirety of the model were instrumented using the Verrou

**Table 2. Naming conventions of instrumented results.**

| Names | Meaning |
|---|---|
| Verificarlo Python | Python Interpreter Instrumented with Verificarlo |
| Verrou Tensorflow/TF | Tensorflow Library Instrumented with Verrou |
| Verrou All | Entire CNN Instrumented with Verrou |

software tool. In order to render the naming conventions clear in the following sections and figures, Table 2 explains the naming schemes used.

We computed the standard deviation, the mean, and the number of significant digits across the MCA samples of (1) classification metrics $S_{min}$, $F_{max}$ and $AUPR$, and (2) class probabilities. $S_{min}$, $F_{max}$ and $AUPR$ were used as metrics as they are the metrics used to evaluate the model in the DeepGOPlus paper [19].

Typical protein function predictors output probabilities that indicate the probability of belonging to a specific function. Therefore, when calculating metrics to measure the predictors' performance, the uncertainty and misinformation linked to the predictions needs to be considered. To this end, the semantic distance between the real and predicted functions based on information content ($S_{min}$) is calculated:

$$S_{min} = \min_{t} \sqrt{ru(t)^2 + mi(t)^2} \tag{6}$$

where $ru(t)$ is the average remaining uncertainty and $mi(t)$ is average misinformation:

$$ru(t) = \frac{1}{n} \sum_{i=1}^{n} \sum_{c \in T_i - P_i(t)} IC(c) \tag{7}$$

$$mi(t) = \frac{1}{n} \sum_{i=1}^{n} \sum_{c \in P_i(t) - T_i} IC(c) \tag{8}$$

where $IC(c)$ is the information content for ontology class $c$, $P_i(t)$ is a set of predicted annotations for a protein $i$ and prediction threshold $t$ and $T_i$ is a set of true annotations, $P_i(t)$.

Moreover, precision and recall are considered and combined into the F-measure to provide one metric. F-measure is the harmonic mean of precision and recall, but to account for thresholding which is often used with protein function predictors, the maximal F-measure over all possible threshold values $F_{max}$ is ultimately used and defined as:

$$F_{max} = \max_{t \in [0.01, 1]} \left\{ \frac{2 \cdot AvgPr(t) \cdot AvgRc(t)}{AvgPr(t) + AvgRc(t)} \right\} \tag{9}$$

where $AvgPr(t)$ is defined as the average precision for a threshold value and $AvgRc(t)$ is the average recall for the same threshold value.

Finally, the usual definition of Area Under the Precision Recall curve ($AUPR$) is used.

## Results

### Sanity check

Table 3 confirms that our installation of DeepGOPlus performs similarly to the original published model. In order to replicate the DeepGOPlus authors' results, the Diamond tool output is included in the protein prediction. Moreover, the various instrumentations of DeepGOPlus

**Table 3. Classification metrics from DeepGOPlus authors (DeepGOPlus), non-perturbed environment (IEEE), MCA-instrumented environments (Verrou TF, Verrou All) and just the CNN itself (DeepGOPlus CNN).** Average values across 10 MCA repetitions for GO Classes MF, CC and BP.

| | Fmax | | | Smin | | | AUPR | | |
|---|---|---|---|---|---|---|---|---|---|
| | **MF** | **CC** | **BP** | **MF** | **CC** | **BP** | **MF** | **CC** | **BP** |
| DeepGOPlus | 0.664 | 0.695 | 0.530 | 9.591 | 10.442 | 44.370 | 0.671 | 0.738 | 0.496 |
| IEEE | 0.6639578 | 0.6944381 | 0.5301602 | 9.5972334 | 10.4480327 | 44.3779111 | 0.6707084 | 0.7379463 | 0.4958606 |
| Verificarlo Python | 0.6639578 | 0.6944381 | 0.5301602 | 9.5972334 | 10.4480327 | 44.3779111 | 0.6707084 | 0.7379463 | 0.4958606 |
| Verrou TF | 0.6639578 | 0.6944381 | 0.5301602 | 9.5972334 | 10.4480327 | 44.3779111 | 0.6707084 | 0.7379463 | 0.4958606 |
| Verrou All | 0.6639578 | 0.6944381 | 0.5301602 | 9.5972334 | 10.4480327 | 44.3779111 | 0.6707084 | 0.7379463 | 0.4958607 |
| DeepGOPlus CNN | 0.4743216 | 0.6580121 | 0.4083806 | 13.164216 | 11.807528 | 49.518513 | 0.4356199 | 0.6847218 | 0.3578813 |

with MCA are also displayed with Diamond included to observe the overall numerical stability of the complete model. All perturbed results are identical to the reference results up to the second digit after the decimal place, after which there is some variation observed. This variation in the order of $10^{-3}$ is to be expected considering potential variations in execution environments between our installation and the authors'. We obtained these results using versions of the DeepGOPlus inference scripts modified to remove variability originating from the Diamond tool and from multithreading. We verified that successive repetitions of the inference with these modifications showed no sign of variability. All the results are obtained with Random Rounding (RR). We also tested PB and full MCA, however, these instrumentation modes caused the model to crash. Finally, the metrics obtained from just running the CNN in an IEEE environment are displayed to serve as the reference against which the MCA instrumentations of just the DeepGOPlus CNN will be compared. This sanity check confirms the validity of our installation and instrumentation of the DeepGOPlus CNN.

Moreover, we confirmed by examining Verrou output that Fused Multiply-Add (FMA) operations are also instrumented, besides regular arithmetic operations. This is essential as FMA operations are key hardware components for training DNNs.

As an additional sanity check, we measure the runtimes and memory consumption of our method, with and without instrumentation to ensure we are not creating an excessive overhead in terms of computational resources. In S2 Appendix, we display these performance metrics in Tables 5 and 6 of S2 Appendix and confirm instrumentation with MCA simply adds a slowdown factor to the project.

## Negligible instabilities found in DeepGOPlus at virtual single-precision 24 bits and virtual double-precision 53 bits

In Fig 3a, we see the standard deviation distribution of protein function class probabilities for different instrumented libraries when noise is introduced with MCA of a magnitude of $10^{-8}$. For the Verrou instrumented Tensorflow library, we see an average standard deviation magnitude of $10^{-8}$ despite some outliers achieving a magnitude of $10^{-7}$. Meanwhile, when all libraries are instrumented with Verrou, we observe a similar distribution of standard deviation among the class probabilities as that of Verrou Tensorflow. However, when instrumenting the Python interpreter with Verificarlo, we see the standard deviation is always zero. No trace of the input noise is seen in the class probabilities of the model as the Python interpreter does not appear to perform many if any of the arithmetic operations used by the model. This instability, represented as standard deviation among the class probabilities, is equivalent to the magnitude of noise being introduced into the DeepGOPlus CNN. Therefore, the magnitude of noise in the output is the same as the magnitude of the noise input to the floating point operations which

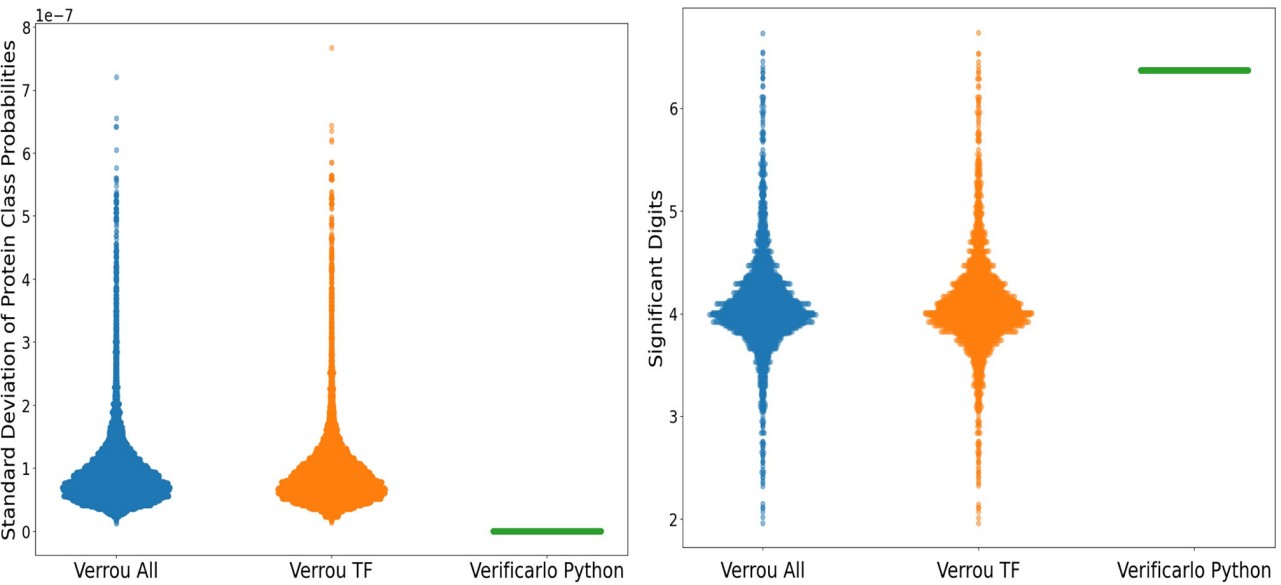

**Fig 3. Variability across protein function class probabilities.** Standard deviations and significant digits were computed on 10 MCA samples, for each of 3,874 proteins. **(A)** Left—Standard Deviation Distribution for Protein Function Class Probabilities **(B)** Right—Distribution of Significant Digits (Verificarlo Python mean: 6.37, Verrou TF mean: 4.08, Verrou All mean: 4.08).

demonstrates there is no internal instability in the model that is amplifying the numerical error. From these results, we can infer that most of the model's arithmetic operations occur within Tensorflow, since all the numerical instability is found in the instrumented Tensorflow library and no instability seems to originate from the instrumented Python interpreter.

The distribution of significant digit values for protein class probabilities in Fig 3b demonstrates that the majority of proteins suffer from no significant information loss from numerical instability. Upon closer examination of the outlier values, these protein class probabilities are very small numbers which makes them more susceptible to the perturbation introduced. Most class probabilities are not small enough to be affected by the perturbation, moreover, low probabilities are filtered out by the thresholding mechanism in the model which ensures the instability is limited in scope.

We note that Verificarlo Python has an average of 6.37 significant digits, while Verrou Tensorflow and Verrou All both have a lower average at 4.08 significant digits. Given that these numbers were calculated in single-precision, this is less than the full 7.23 significant digits available, especially in the case of Verrou Tensorflow and Verrou All, which are both more unstable than Verificarlo Python. However, since 4 significant digits is still sufficient to have reliable protein class probabilities, this drop is considered negligible.

As shown in Table 4, the classification metrics have an average of 6.27 significant digits. Moreover, we observe that there is only consistent variability among the $AUPR$ values which have a standard deviation at a magnitude of $10^{-7}$. However, for the rest of the classification metrics, the results do not vary from the IEEE classification metric values obtained and, in fact, the variability found in $AUPR$ is ultimately negligible.

This difference in metric stability might be explained by the fact that $AUPR$ is calculated by taking the precision and recall calculated at each threshold value between 0 and 1 while the $F_{max}$ and $S_{min}$ displayed are always the metrics found after finding the optimal threshold value. However, another factor that might also be impacting AUPR's stability is the fact that it is

**Table 4. DeepGOPlus metrics and significant digits across instrumented environments.**

| | | Fmax | | | Smin | | | AUPR | | |
|---|---|---|---|---|---|---|---|---|---|---|
| | | Mean | Standard Deviation | Significant Digits | Mean | Standard Deviation | Significant Digits | Mean | Standard Deviation | Significant Digits |
| Verificarlo Python | MF | 0.4743216 | 0.00 | 6.37 | 13.164216 | 0.00 | 6.37 | 0.4356199 | 0.00 | 6.37 |
| | CC | 0.6580121 | 0.00 | 6.37 | 11.807528 | 0.00 | 6.37 | 0.6847218 | 0.00 | 6.37 |
| | BP | 0.4083806 | 0.00 | 6.37 | 49.518513 | 0.00 | 6.37 | 0.3578813 | 0.00 | 6.37 |
| Verrou TF | MF | 0.4743216 | 0.00 | 6.37 | 13.164216 | 0.00 | 6.37 | 0.4356191 | 1.1455358e- 6 | 5.18 |
| | CC | 0.6580121 | 0.00 | 6.37 | 11.807528 | 0.00 | 6.37 | 0.6847219 | 8.9023116e- 8 | 24 |
| | BP | 0.4083806 | 0.00 | 6.67 | 49.518513 | 0.00 | 6.37 | 0.3578815 | 1.1737338e- 7 | 5.94 |
| Verrou All | MF | 0.4743216 | 0.00 | 6.37 | 13.164216 | 0.00 | 6.37 | 0.4356194 | 7.7922517e- 6 | 5.35 |
| | CC | 0.6580121 | 0.00 | 6.37 | 11.807528 | 0.00 | 6.37 | 0.6847218 | 5.4863019e- 8 | 6.42 |
| | BP | 0.4083806 | 3.2739540e- 8 | 6.71 | 49.518513 | 0.00 | 6.37 | 0.3578815 | 6.7939657e- 8 | 6.18 |

calculated using a numerical integration with a trapezoidal scheme. Doing so involves some level of approximation of the shape of the AUPR curve, which could also factor into the instability found. Nonetheless, the calculation is known to be considerably stable and would only introduce noise in the magnitude of $10^{-16}$, which is much smaller than the magnitude found suggesting the instability found is from the perturbed results.

## Selective reduction of precision of DeepGOPlus inference

Given the stability of the DeepGOPlus model, this opens up the possibility of exploring reduced precision formats for the DeepGOPlus model with VPREC. To this end, the float32, float16, bfloat16 and bfloat8 low-precision numerical formats are tested in order to provide benchmarks of the DeepGOPlus model's performance at each reduced format. Moreover, these formats are implemented using inbound and outbound modes. Outbound mode reduces the output of the arithmetic operations to the chosen reduced precision while inbound mode leaves the operation calculations at full-precision while reducing the input values. Given that both the double and single precision numbers are used by DeepGOPlus, sixteen different combinations of the above four reduced formats are tested and displayed in Fig 4a and 4b. In these figures we present the results for the $AUPR$ metric, but the results for $F_{max}$ and $S_{min}$ can be found in Figs 6 and 7 of S1 Appendix. We visualize the impact reduced precision has on the model's performance by measuring the relative difference between metrics obtained with reduced precision and those obtained at full precision.

When examining the performance of the numerical formats in Fig 4a, there is a substantive drop in performance with an average of -2% for $F_{max}$, -14% for $AUPR$ and -20% for $S_{min}$ across all protein classes. Meanwhile, for Fig 4b, we observe that any combination of reduced double-precision with full single-precision for inbound mode achieves comparable performance to the full IEEE metrics. We also note, there is an average drop of -6% for $F_{max}$, -13% for $AUPR$ and -12% for $S_{min}$ (excluding the format combinations that achieve comparable performance), which seems to indicate that inbound mode has better general performance than outbound mode. We observe inbound mode achieving comparable performance whenever double-precision is reduced, but single-precision is left intact. This is unlike the results observed in outbound mode, where despite $F_{max}$ occasionally surpassing the full IEEE metrics, the overall performance decreases. Besides supporting the idea that inbound mode achieves better performance, this implies a core arithmetic operation is occurring in double-precision where the calculations cannot be reduced as is being done in outbound mode. We also observe that with

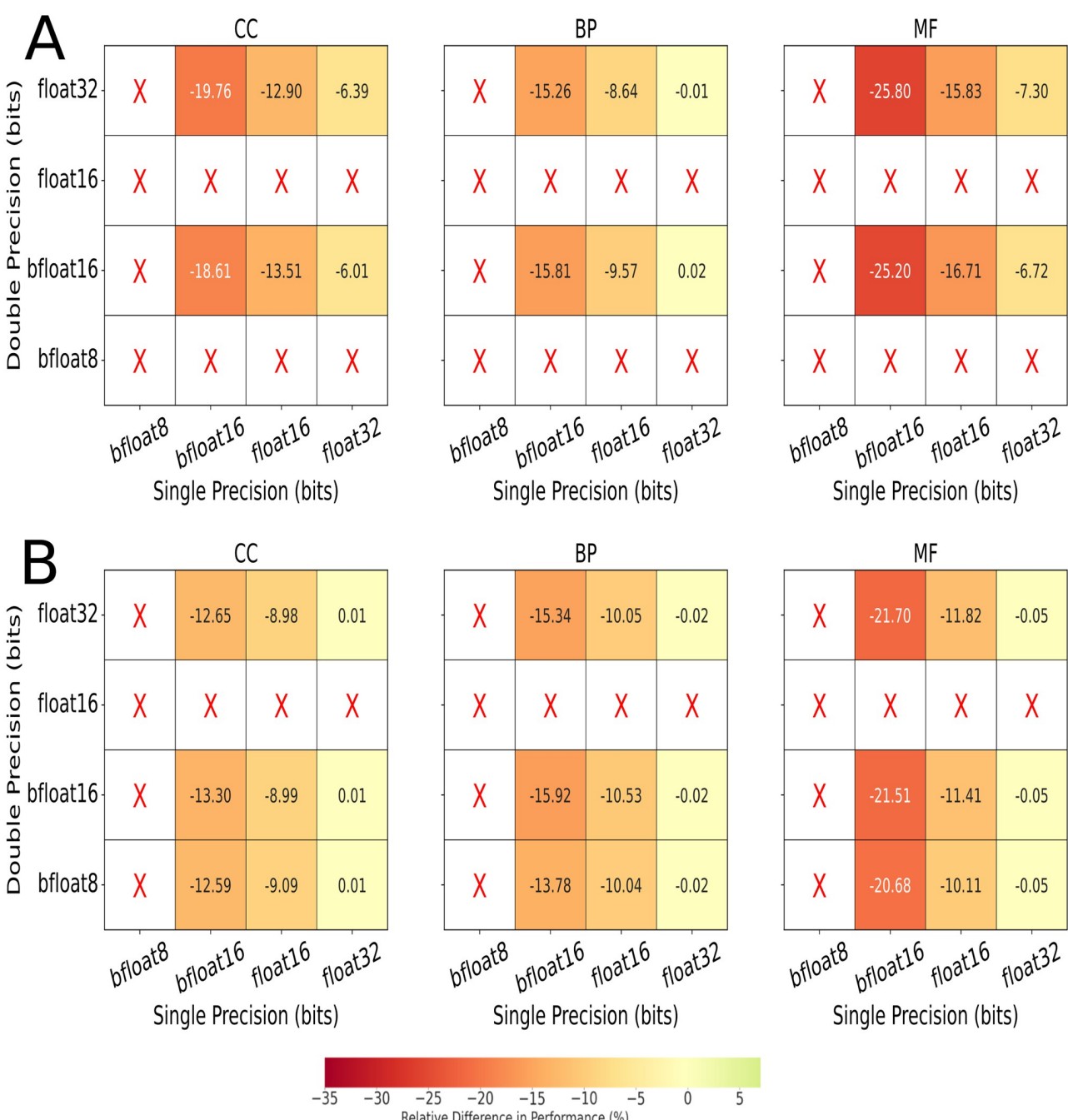

**Fig 4. Difference in AUPR Performance between Reduced Precision Formats and IEEE for GO Classes Using Outbound (A) and Inbound Precision (B) Only.** The red X signifies the model crashed at these precision values. As all metrics experienced similar effects only AUPR is shown, while the other metric heatmaps can be found in Figs 6 and 7 of S1 Appendix.

inbound mode, double-precision can be reduced to bfloat8 while with outbound mode, only single-precision and bfloat16 are achievable formats. Interestingly enough, in both cases, setting double-precision to float16 causes Tensorflow to abort upon starting up while setting single-precision to bfloat8 only throws a calculation error once the model has begun running.

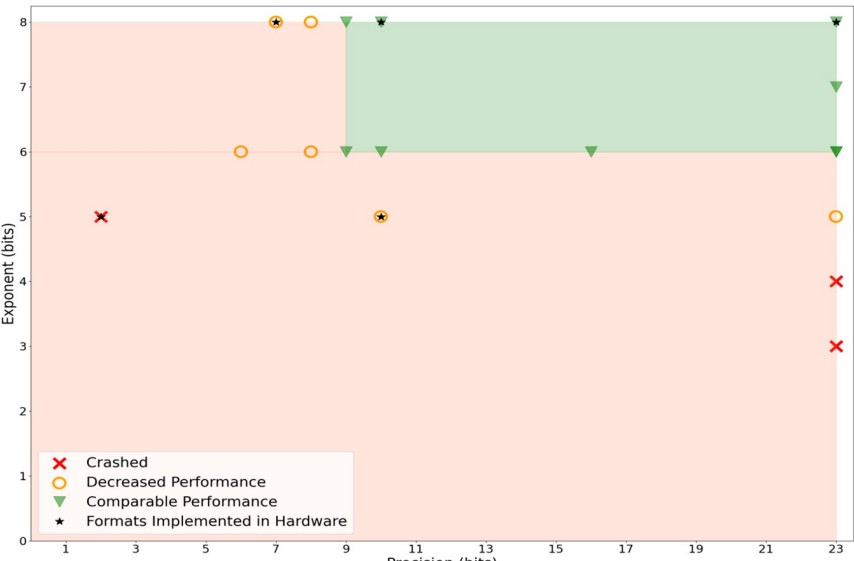

**Fig 5. Search for optimal precision and exponent values before performance drop off for VPREC inbound mode in single-precision.** Double-precision is untouched and the green shaded region marks the area of acceptable values for reduced precision and comparable performance. Fig 8 of S1 Appendix demonstrates these findings hold when double-precision is reduced.

This implies that double-precision is necessary in ensuring Tensorflow, the software running the model, works correctly. Considering most of the arithmetic operations occur in single-precision, the model crashing due to setting single-precision to bfloat8 can perhaps be attributed to numerical overflow/underflow. While studies have utilised bfloat16 formats with DNNs, the usual approach is leaving the calculations to be performed in float32 and selectively using the bfloat16 format to store intermediate computations and the data input to the model. In this case, outbound precision mode reduces all operations to bfloat8 indiscriminately. Nonetheless, while inbound precision mode should be leaving most operations at full-precision, it must still be reducing some essential ones, thus explaining why the model crashed, despite bfloat8 being used successfully in other scenarios.

Given inbound mode's better performance and the overall reliance on single-precision, Fig 5 identifies at which point DeepGOPlus' performance degrades when single-precision is reduced and double-precision is left intact. Both the exponent and the precision components of single-precision values were explored to find the lowest format before performance begins deteriorating. The minimum acceptable format found was composed of 9 bit precision and 6 bit range. We observed that precision could be reduced to more than half of its original state without a negative impact, but the exponent could only be reduced a maximum of 2 bits. We observe the same results when double-precision is reduced to single-precision as is shown in Fig 8 of S1 Appendix. Nonetheless, there is currently only one available hardware implementation for the reduced precision formats that were successful which is TensorFloat 32 (TF32) [55]. TF32 is composed of an exponent of 8 bits and a precision of 10 bits and is currently only implemented in Nvidia A100 GPUs. The limited availability of this format restricts the implementation of reduced precision in DeepGOPlus for end-users, but is a promising start.

## Discussion

### DeepGOPlus numerical stability

The DeepGOPlus CNN is numerically stable as negligible variation is observed among the classification metrics as a result of MCA perturbations with double-precision values at precision 53 bits and single-precision values at precision 24 bits. The variation observed in the predicted class probabilities in response to this perturbation is on average of a magnitude of $10^{-8}$—equivalent to the magnitude of the applied numerical perturbations—which indicates high numerical stability. The small variations induced through MCA carries to the performance metrics used for evaluation (e.g., precision, recall). However the effect is minimal as the resulting variability is at most a magnitude $10^{-6}$ which doesn't change the overall performance. Moreover, the $F_{max}$ and $S_{min}$ metrics involve the use of the max and min operations, which are demonstrated to have a regularizing effect in our case, contrary to $AUPR$, which was the only metric where consistent variability was observed.

The fact that inference through DeepGOPlus is numerically stable implies that it can be executed in different environments without any expected variations. This means that users of DeepGOPlus can depend on the pre-trained model made available to predict results that are reproducible.

Therefore, previous concerns about the numerical stability of CNNs as well as other DNNs from the literature [29–32] do not manifest in DeepGOPlus inference, which may be due to several reasons. First, the relatively shallow architecture of the DeepGOPlus CNN model may favor numerical stability. Indeed, even though the DeepGOPlus model contains approximately 60 million trainable parameters, which is within the common range for a CNN, it only has 3 layers (convolution, max-pooling, fully connected), which limits the number of consecutive floating-point operations done during inference and therefore limits the numerical noise that can occur within the model.

The observed stability may similarly result from the relative simplicity of the data being input to the model. Protein sequences typically consist of 50–2000 amino acids, while other types of data commonly processed through CNNs, such as images, can regularly be composed of hundreds of thousands of elements. As a result, DeepGOPlus inference requires less floating point arithmetic operations than models that deal with more complex data, which favors numerical stability.

Adversarial attacks have not been widely documented in protein classification, especially relative to their frequency in image classification, and the observed numerical stability of DeepGOPlus inference does not imply its immunity to potential adversarial attacks. Adversarial attacks involve data perturbations which are different in nature from the numerical perturbations studied in our experiments. However, in future work it will be interesting to study if the model's current stability extends to withstanding adversarial attacks. If so, further exploration into whether this trait is due to DeepGOPlus' architecture or is a property of the data itself would be illuminating.

### Necessity for selective reduced precision

Our reduced precision experiments have demonstrated that precision should only be reduced in inbound mode in order to achieve comparable performance. Single-precision can only be reduced to one format with an existing hardware implementation, while double-precision can be reduced bfloat16. We observed that when double-precision numbers are reduced in outbound mode, the performance of the model dropped drastically, while in inbound mode, the precision can be halved or reduced even further while still obtaining comparable results. This

implies that a core arithmetic operation is being calculated in double precision and it cannot have its operations reduced, only its input values. Upon closer examination, this arithmetic operation has been traced to Numpy's multiarray_umath module which contains operations like matrix dot products. Although the model predominantly uses single-precision, double-precision is still present in crucial arithmetic operations, potentially affecting overall performance.

Given that single-precision is used for the remaining 99% of all instrumented operations, reducing it would have the most impact on memory and CPU consumption. Leveraging Deep-GOPlus's numerical stability, we successfully reduced single-precision to TensorFloat 32 while maintaining comparable performance. However, it is essential to note that TensorFloat 32 is currently supported only by Nvidia A100 GPUs. For users without this GPU, double-precision numbers could be either reduced in inbound mode, or the Numpy multiarray_umath module could be excluded from reduction, all while keeping the single-precision numbers at float32 in order to achieve comparable results. However, given the limited presence of double-precision in the DeepGOPlus CNN, even selective reduced precision would result in minimal impact on the memory and CPU consumption of the model.

Another takeaway from this experiment is the performance of bfloat16 compared to float16. Bfloat16 could be utilised for double-precision without issues, while float16 for double-precision sometimes led to model crashes. One might initially assume that the model is unable to run float16 for double-precision numbers do to its smaller exponent size, but keeping in mind that for VPREC inbound mode, bfloat8, which has an equally small exponent works successfully, this theory does not hold. When looking over Tensorflow documentation, we see recommendations to carefully implement float16 and many Github issues exist involving float16 implementations in Tensorflow models. Therefore the float16 model crashes are possibly due to compatibility challenges between float16 and TensorFlow, which underscores the need for careful implementation of float16.

Surprisingly, when single-precision was reduced to float16 and bfloat16, we see that the classification performance is much better with float16, despite how ill-suited it is for double-precision. In fact, with float16 in single-precision, we observe an average performance drop of 6% across all metrics, while bfloat16 has an average drop of 15%. Therefore, while bfloat16 has a larger dynamic exponent, the model appears to require more precision when doing floating point operations which is provided by float16. In brief, double and single-precision are being used for different purposes in this model and have different requirements in terms of precision and exponent sizes, which need to be taken into account prior to reducing the numerical precision formats in order to optimize performance.

This result might be explained by the fact the DeepGOPlus model was pre-trained for single and double precision. Using reduced precision during training might produce models supporting reduced precision during inference as well. Indeed, perhaps not all bits in the single-precision mantissa are useful but as the model is currently trained to utilise them all, future work might involve re-training it to use less. There are several examples in the literature [18] that demonstrate deep learning algorithms including CNNs can achieve high performance with reduced precisions.

## Verrou & Verificarlo comparison

Both Verrou and Verificarlo are tools for numerical uncertainty quantification and do so by implementing MCA. Moreover, both were set to use virtual precision 24 for single-precision and virtual precision 53 for double-precision in order to be consistent across the numerical instability quantification. However, they differ in their implementation. As a whole, Verrou can be used with programs across languages with no instrumentation, unlike Verificarlo

which requires the programs targeted to be recompiled. Nonetheless, we observed Verrou can run up to approximately 120x slower than Verificarlo, which can become a bottleneck in any project if not utilised carefully.

As an additional sanity check, the numerical uncertainty quantification results obtained from Verificarlo and Verrou were compared. In order to have a complete comparison, results would be required from the Tensorflow library instrumented by Verificarlo, but this was not possible due to compatibility issues with the clang compiler. However, we can compare the results obtained from instrumenting the Python interpreter with both Verrou and Verificarlo. When examining Table 3, we see that in both cases, instrumenting Python does not result in any instability. We see that when Python is instrumented alone with Verificarlo, no instability is found. Meanwhile, when Python is instrumented with Tensorflow by Verrou, it appears that Tensorflow is the sole source of instability. Therefore, we can confirm that for DeepGOPlus, the results obtained with either Verificarlo or Verrou are comparable to one another and that the tools can be used interchangeably at one's discretion.

## Conclusion

This paper determined the numerical stability of the DeepGOPlus inference model, focusing on its CNN component built to predict protein function from protein sequences. To this end, we instrumented DeepGOPlus using Monte Carlo Arithmetic (MCA) to inject numerical noise into the model and measure the resulting variability in protein class probabilities and classification metrics. We found DeepGOPlus inference to be highly numerically stable, which should ensure the reproducibility and robustness of DeepGOPlus predictions across execution environments.

In addition, we evaluated the performance of the DeepGOPlus inference stage with reduced precision formats. Notably, we observed that reducing precision in outbound mode caused larger performance drops across all tested formats compared to inbound mode. This underscores the significance of assigning adequate precision to arithmetic operations for their calculations. While some core arithmetic operations in the inference stage could be reduced to lower precision formats, limitations arose, particularly concerning the exponent component. This restriction restricts the available hardware-implementable formats, leaving TensorFloat 32 as the sole available format for single-precision in DeepGOPlus inference, available only on select Nvidia GPUs. In conclusion, while the DeepGOPlus inference stage is stable in its current state, most arithmetic operations cannot have their precision reduced without suffering a substantive drop in performance.

Generalizing these findings to other CNNs presents challenges due to the diversity of possible implementations that can impact numerical stability. Common architectural features like pooling operations may help reduce numerical uncertainty, but software tools and input data types can significantly affect a model's numerical stability. Therefore, further experimental investigations into the stability of different CNNs [56], as well as theoretical explorations of CNN architecture stability are needed to establish more robust generalisations about CNNs.

## Future work

Future work could study the numerical stability and potential use of reduced precision in the training step of the DeepGOPlus model as this step is an important determinant of DeepGO-Plus' performance. However, training the DeepGOPlus model with MCA is especially compute intensive. Additional overhead can be expected due to the MCA instrumentation as MCA does not currently support multi-threading and can only function on CPU. Addressing the computational intensity of MCA-instrumented training could involve strategies such as

implementing distributed training through data parallelism. Considering mixed precision in addition to reduced precision would also be relevant, as some layers might require higher precision than other ones [15–17]. Furthermore, studying deeper CNNs than DeepGOPlus, where the risk of exploding or vanishing gradients [57] is prevalent and may affect numerical stability, could yield insights into the numerical properties of CNNs in a broader context. Owing to the widespread adoption of CNNs in various domains, the practice of employing MCA during CNN inference to investigate numerical stability has extended beyond its original application to DeepGOPlus and can now be used in other fields, including medical imaging [56].

## Supporting information

**S1 Appendix. Performance at reduced precision.**
(PDF)

**S2 Appendix. Performance metrics with and without instrumentation.**
(PDF)

## Acknowledgments

Special thanks are extended to the authors of the DeepGOPlus paper and for the support they provided answering queries about their work. Special thanks are also extended to Mathieu Dugré, for his contribution regarding reduced precision formats' hardware implementation, particularly regarding TensorFloat 32. Computations were made on the Narval and Béluga supercomputers from École de Technologie Supérieure (ETS, Montréal), managed by Calcul Québec and Compute Canada. The operation of these supercomputers are funded by the Canada Foundation for Innovation (CFI), le Ministère de l'Économie, des Sciences et de l'Innovation du Québec (MESI) and le Fonds de recherche du Québec—Nature et technologies (FRQ-NT). [58]

## Author Contributions

**Investigation:** Inés Gonzalez Pepe.

**Methodology:** Inés Gonzalez Pepe, Yohan Chatelain.

**Resources:** Tristan Glatard.

**Software:** Yohan Chatelain.

**Supervision:** Tristan Glatard.

**Validation:** Yohan Chatelain, Gregory Kiar, Tristan Glatard.

**Writing – original draft:** Inés Gonzalez Pepe.

**Writing – review & editing:** Inés Gonzalez Pepe, Yohan Chatelain, Gregory Kiar, Tristan Glatard.

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
