## [Decision Letter · Decision Letter 0]

29 May 2023

PONE-D-23-12824Numerical Stability of DeepGOPlus InferencePLOS ONE

Dear Dr. Gonzalez Pepe,

Thank you for submitting your manuscript to PLOS ONE. After careful consideration, we feel that it has merit but does not fully meet PLOS ONE’s publication criteria as it currently stands. Therefore, we invite you to submit a revised version of the manuscript that addresses the points raised during the review process.

We look forward to receiving your revised manuscript.

Kind regards,

Abdullah Almuhaimeed

Academic Editor

PLOS ONE

Journal Requirements:

Additional Editor Comments:

Dear Authors,

Your manuscript is worthy investigated. However, after revising your manuscript we found some major issues that need to be revised. Please check reviewers comments when resubmitting your work.

Regards

Reviewers' comments:

Reviewer's Responses to Questions

**Comments to the Author**

1. Is the manuscript technically sound, and do the data support the conclusions?

Reviewer #1: Partly

Reviewer #2: Yes

Reviewer #3: Partly

2. Has the statistical analysis been performed appropriately and rigorously? 

Reviewer #1: Yes

Reviewer #2: Yes

Reviewer #3: Yes

3. Have the authors made all data underlying the findings in their manuscript fully available?

Reviewer #1: Yes

Reviewer #2: Yes

Reviewer #3: Yes

4. Is the manuscript presented in an intelligible fashion and written in standard English?

Reviewer #1: No

Reviewer #2: Yes

Reviewer #3: Yes

5. Review Comments to the Author

Reviewer #1: This paper investigates the opportunity to use reduced-precision floating point formats for DeepGOPlus inference to reduce memory consumption and latency. This is meaningful. Howerver, the current version of the paper is poorly readable.

1. It's recommended to add an overall structure diagram to highlight the innovative points and illustrate the network architecture of the paper.

2. The author should improve the format and language (such as formula numbering, etc.) of the paper to increase its readability.

Reviewer #2: The article presents a nice approach that demonstrates the numerical stability of a state-of-the-art sequence-based protein function prediction method named ‘DeepGOPlus’. Although the approach is highly interesting there are multiple issues that need to be resolved before the article can be considered for publication in the journal.

Major Issues

1. The approach discussed in the manuscript can be applied to any CNN-based deep learning method for a certain task. The motivation and rationale behind the choice of ‘sequence-based protein function prediction’ as the task is not clear. This obscures the main theme of the article, that in turn confuses the reader, who doesn’t get to understand what is the main message of this article? Please rewrite the ’Abstract’ and ‘Introduction’ sections of the manuscript in a way that this confusion doesn’t arise.

2. What are some other methods to achieve the two goals discussed in the 3rd paragraph of page 2? What is reason behind choosing MCA and not these other methods? Please explain.

Reviewer #3: Overall, the study presented raises important concerns and provides valuable insights into the numerical stability and precision requirements of DeepGOPlus, a CNN used for predicting protein function. Here are some major concerns on the study:

1. The paper mentions investigating the opportunity to use reduced-precision floating point formats for DeepGOPlus inference, but it does not thoroughly discuss the potential impact of reduced precision on the model's performance and prediction accuracy. The study should have provided a more detailed analysis of how the reduced precision affects the model's output quality and whether any trade-offs exist between precision reduction and prediction reliability.

2. The study focuses solely on the inference stage of DeepGOPlus, which is justified based on its importance and likelihood of exposure to noise. However, it would have been beneficial to explore the impact of reduced-precision formats on the model's training phase as well. Training is a critical stage where accuracy and stability are equally important, and understanding the effects of precision reduction during training would provide a more comprehensive analysis.

3. The study employs Monte Carlo Arithmetic and VPREC to analyze the impact of reduced-precision formats. While these are valid techniques, it would have been valuable to compare their results against alternative precision reduction approaches, such as quantization or low-bit representations. Such comparisons would help establish the relative effectiveness and limitations of different precision reduction methods.

4. The paper briefly mentions numerical stability challenges in DNNs and their sensitivity to noise injection, but it does not elaborate on the specific challenges encountered during the investigation of DeepGOPlus. Providing more details and discussing the implications of these challenges on the model's performance and reliability would enhance the study's contribution.

5. The study concludes that DeepGOPlus cannot currently be implemented with a lower-precision floating-point format. However, the paper does not provide quantitative performance metrics or an extensive evaluation of the model's accuracy, computational efficiency, and memory consumption under different precision formats. Including such metrics and a comprehensive evaluation would strengthen the conclusions and support the study's claims effectively.

6. The study focuses on DeepGOPlus, and its findings may not directly generalize to other CNN architectures. It would be valuable to discuss the implications of the study's results and insights in a broader context, addressing how the observed numerical stability and precision requirements may vary for different CNN models used in diverse domains.

7. Deep learning is well-known and has been used in previous bioinformatics studies i.e., PMID: 36166351, PMID: 34730875. Therefore, the authors are suggested to refer to more works in this description to attract a broader readership.

8. The paper provides a link to the project's code repository, which is commendable. However, it would be beneficial to have a web server to represent their model. This would enhance the reproducibility and enable interested researchers to build upon the work.

9. The authors should compare their performance results to previously published works on the same problem/data.

6. PLOS authors have the option to publish the peer review history of their article (what does this mean?). If published, this will include your full peer review and any attached files.

Reviewer #1: No

Reviewer #2: No

Reviewer #3: No

---

## [Author Response · Author response to Decision Letter 0]

25 Sep 2023

Dear Editor and Reviewers,

We would like to thank you again for considering our manuscript for publication in PLOS One and for the thorough review. Please find below the answer to your comments. The corresponding revision of our manuscript is attached, where revisions are highlighted in blue.

We thank you again for your time and consideration, and we remain available for any further information you may need.

Yours sincerely,

Inés Gonzalez Pepe, on behalf of the authors

---

## [Decision Letter · Decision Letter 1]

3 Nov 2023

PONE-D-23-12824R1Numerical Stability of DeepGOPlus InferencePLOS ONE

Dear Dr. Gonzalez Pepe,

Thank you for submitting your manuscript to PLOS ONE. After careful consideration, we feel that it has merit but does not fully meet PLOS ONE’s publication criteria as it currently stands. Therefore, we invite you to submit a revised version of the manuscript that addresses the points raised during the review process.

We look forward to receiving your revised manuscript.

Kind regards,

Kathiravan Srinivasan

Academic Editor

PLOS ONE

**Additional Editor Comments:**

Please revise and resubmit your manuscript. 

Reviewers' comments:

Reviewer's Responses to Questions

**Comments to the Author**

1. If the authors have adequately addressed your comments raised in a previous round of review and you feel that this manuscript is now acceptable for publication, you may indicate that here to bypass the “Comments to the Author” section, enter your conflict of interest statement in the “Confidential to Editor” section, and submit your "Accept" recommendation.

Reviewer #3: All comments have been addressed

Reviewer #4: (No Response)

Reviewer #5: All comments have been addressed

Reviewer #6: (No Response)

Reviewer #7: All comments have been addressed

2. Is the manuscript technically sound, and do the data support the conclusions?

Reviewer #3: Yes

Reviewer #4: Yes

Reviewer #5: Yes

Reviewer #6: Yes

Reviewer #7: No

3. Has the statistical analysis been performed appropriately and rigorously? 

Reviewer #3: I Don't Know

Reviewer #4: Yes

Reviewer #5: Yes

Reviewer #6: Yes

Reviewer #7: No

4. Have the authors made all data underlying the findings in their manuscript fully available?

Reviewer #3: Yes

Reviewer #4: Yes

Reviewer #5: Yes

Reviewer #6: Yes

Reviewer #7: No

5. Is the manuscript presented in an intelligible fashion and written in standard English?

Reviewer #3: Yes

Reviewer #4: Yes

Reviewer #5: Yes

Reviewer #6: Yes

Reviewer #7: Yes

6. Review Comments to the Author

Reviewer #3: My previous comments have been addressed. The manuscript has been improved and can meet the requirement for publication.

Reviewer #4: Numerical Stability of DeepGOPlus Inference

My comments and suggestions are the following:

1.What are the three main advantages and drawbacks of DeepGOPlus as compared to the existing methods?

2.What are the key benefits of “DeepGOPlus CNN” model as compared to “HGANet” and “CSLNet” models which were presented in these research work (“Automated identification of human gastrointestinal tract abnormalities based on deep convolutional neural network with endoscopic images”) and (“Automated multi-class classification of skin lesions through deep convolutional neural network with dermoscopic images”). Explain briefly main benefits of DeepGOPlus CNN over HGANet and CSLNet in Introduction section.

3.It is suggested to authors to make a detailed block diagram or flowchart of proposed method so that the readers can easily understand it.

4.It is suggested to the authors to use officials email address instead of “@gmail.com”.

5.To establish the numerical stability of DeepGOPlus, authors quantified the numerical uncertainty of floating point model using Monte Carlo Arithmetic. Are there other techniques instead of Monte Carlo?

Reviewer #5: In my estimation, you have largely succeeded in improving the manuscript in accordance with the reviewers' comments.

The article now reads more comprehensive and well-runded overall.

In light of its strengths, primarily relevance, novelty and thoroughness in terms of its stated objective, I am hereby recommending approval for publication.

NOW THE PAPER IS SUITABLE FOR PUBBLICATION

Reviewer #6: The article delves into a fundamental aspect for the generation of robust and reliable deep learning models: the evaluation of numerical stability. As the authors rightly point out, this is particularly important in the case of determining a protein's function via deep learning. This evaluation assesses how resilient a model is when subjected to minor numerical perturbations during calculations. The article is well-researched, with a detailed analysis and the application of various verification methods. In terms of its structure, it is comprehensive, clear, and instructive. Overall, it is a well-executed article with intriguing and thought-provoking results. I want to emphasize the diligent work of the authors.

I would like to offer a few minor suggestions to enhance the understanding of certain aspects of the article and provide some observations regarding minor structural aspects of the article.

-In my personal opinion, I would remove the quote "The code for this project can be found at " ext-link-type="uri" xlink:type="simple">https://github.com/big-data-lab-team/deepgoplus-stability" from the abstract and place it in a specific section, but not in the abstract.

-Probably, I would consolidate all the links to data and code used in the article in a separate section, regardless of whether access is referenced in different sections.

-Around line 30 ("This paper centers its investigation on DeepGOPlus [19], a state-of-the-art CNN protein function classification model. It is chosen due to being one of the top performers in the CAFA3 challenge [20], a community-wide challenge to assess computational protein function prediction methods."), it might be desirable to delve further into the selection of DeepGoPlus as the model of choice. An explanation in this regard or a reference to more recent developments could be interesting, including a more quantitatively substantiated justification. Please consider adding one or more paragraphs where you compare it to these alternatives and the added value of your choice.

-It would be interesting to include in DeepGOPlus Model Section an image that provides a schematic of the architecture of the DNN model used. Perhaps, placing it around line 286 would be beneficial. This would assist the reader in understanding the network's structure.

-A "Future Work" section that compiles the future phases of this research would be beneficial. For instance, the content from line 620 onwards is likely better placed in a dedicated section rather than within the "Conclusions”.

Reviewer #7: 1. The authors must provide statistical values related to DeepGOPlus inference time before and after reduced precision floating point formats, memory consumption, and latency values. Provide all the details in a Table format.

2. How the current work is novel and highlight the contributions.

3. There are several LSTM techniques for sequence-based predictions; authors must justify the reason for considering CNN suitable for classification tasks.

4. Authors must mention the computational complexity of the model related to state-of-the-art techniques with hardware specifications such as GPU RAM system configurations.

5. Authors have just combined a few existing techniques. Is it sufficient for a standard journal? Please justify the same.

7. PLOS authors have the option to publish the peer review history of their article (what does this mean?). If published, this will include your full peer review and any attached files.

Reviewer #3: No

Reviewer #4: No

Reviewer #5: **Yes: **GIUSEPPE GULLO

Reviewer #6: **Yes: **Ana Guerrero-Tamayo

Reviewer #7: **Yes: **Jagan Mohan N

---

## [Author Response · Author response to Decision Letter 1]

30 Nov 2023

Responses are provided in the attached Response to Reviewers PDF

---

## [Decision Letter · Decision Letter 2]

18 Dec 2023

Numerical Stability of DeepGOPlus Inference

PONE-D-23-12824R2

Dear Dr. Gonzalez Pepe,

We’re pleased to inform you that your manuscript has been judged scientifically suitable for publication and will be formally accepted for publication once it meets all outstanding technical requirements.

Kind regards,

Kathiravan Srinivasan

Academic Editor

PLOS ONE

Additional Editor Comments (optional):

Reviewers' comments:

Reviewer's Responses to Questions

**Comments to the Author**

1. If the authors have adequately addressed your comments raised in a previous round of review and you feel that this manuscript is now acceptable for publication, you may indicate that here to bypass the “Comments to the Author” section, enter your conflict of interest statement in the “Confidential to Editor” section, and submit your "Accept" recommendation.

Reviewer #6: All comments have been addressed

Reviewer #7: All comments have been addressed

2. Is the manuscript technically sound, and do the data support the conclusions?

Reviewer #6: Yes

Reviewer #7: Partly

3. Has the statistical analysis been performed appropriately and rigorously? 

Reviewer #6: Yes

Reviewer #7: Yes

4. Have the authors made all data underlying the findings in their manuscript fully available?

Reviewer #6: Yes

Reviewer #7: Yes

5. Is the manuscript presented in an intelligible fashion and written in standard English?

Reviewer #6: Yes

Reviewer #7: No

6. Review Comments to the Author

Reviewer #6: Each of my comments has been accurately reflected. From my perspective, the article should be published.

Reviewer #7: The authors have addressed all the questions and the quality of the manuscript is now improved for publication.

7. PLOS authors have the option to publish the peer review history of their article (what does this mean?). If published, this will include your full peer review and any attached files.

Reviewer #6: **Yes: **Ana Guerrero-Tamayo

Reviewer #7: **Yes: **N Jagan Mohan

---

## [Editor Report · Acceptance letter]

18 Jan 2024

PONE-D-23-12824R2 

PLOS ONE

Dear Dr. Gonzalez Pepe, 

I'm pleased to inform you that your manuscript has been deemed suitable for publication in PLOS ONE. Congratulations! Your manuscript is now being handed over to our production team.

Kind regards, 

on behalf of

Dr. Kathiravan Srinivasan 

Academic Editor

PLOS ONE